# Analysis of Research on the SDGs: The Relationship between Climate Change, Poverty and Inequality

**María del Carmen Pérez-Peña, Mercedes Jiménez-García** **, Jose Ruiz-Chico \*** and **Antonio Rafael Peña-Sánchez**

INDESS (University Institute for Sustainable Social Development), Department of General Economy, Faculty of Social Sciences and Communication, University of Cadiz, Avda. de la Universidad, 4-11406 Jerez de la Frontera, Spain; carmen.perezpena@uca.es (M.d.C.P.-P.); mercedes.jimenezgarcia@uca.es (M.J.-G.); rafael.pena@uca.es (A.R.P.-S.)
\* Correspondence: jose.ruizchico@uca.es

**Abstract:** Since its adoption in September 2015, the 2030 Agenda has laid the foundations for a model of shared prosperity for a sustainable world. The current global pandemic highlights profound inequalities affecting our economies, health, and quality of life. For this reason, the aim of this study was to present the current state of scientific research related to inequality, poverty, and climate change, and to propose lines of improvement that can contribute to achieving three of the 17 SDGs (end poverty, SDG 1; reduce inequality, SDG 10; and climate action, SDG 13), proposed in the 2030 Agenda. For this purpose, we undertook a systematic literature review. The results show that the subject of poverty, inequality, and climate change has been little studied or articulated by researchers, and significant differences exist between the different areas studied. The highest number of publications (51.7%) is associated with topics related to sustainability—environment and economics. The remainder are distributed among 12 other research areas. Another relevant finding is that the effects of climate change are more pressing for more vulnerable populations, including impoverished women from rural areas and children from underdeveloped countries. This gender and social inequality has been rarely addressed in studies. Food security and energy poverty is another under-researched area. According to the results obtained in this work, we consider that the circular economy may improve these indicators, constituting a line of future research. Thus, one of the main objectives of this approach is to eliminate negative externalities, specifically the existing social inequalities within the current linear economy model.

**Keywords:** food security; agriculture; health; sustainable development goals; 2030 agenda; gender studies; COVID-19



## 1. Introduction

The 2030 Agenda sets out 17 goals with 169 inclusive and indivisible targets covering the economic, social, and environmental spheres. Their main aim is to ensure good livelihoods, free from poverty and hunger, in healthy and safe environments. They also aim to successfully combat the threats of climate change with sustainable production patterns and efficient and effective economies [1]. These 17 goals include ending poverty (SDG 1), reducing inequalities (SDG 10), and climate action (SDG 13) [2].

Extreme poverty has most often been associated with rural areas [3]. Globally, 78% of people living in extreme poverty live in such an environment and depend on agriculture for their livelihoods [4]. In 2018, four of five people affected by poverty lived in rural areas, with children and women being most affected. The areas most impacted by this situation are concentrated in Southern Africa and India [5,6]. However, due to the COVID-19 pandemic, extreme poverty is spreading to overcrowded urban centres, and thus affecting a population that depends on informal and manufacturing services for its livelihood. In the face of this paradigm shift, extreme poverty will affect more than 150 million people in 2021, who will have to subsist on less than USD 1.90 a day [7]. This increase in extreme

poverty is among the effects associated with the loss of employment and the decrease in income and salaries resulting from COVID-19. As a consequence of this situation, the most impoverished families in Latin America and sub-Saharan Africa are being forced to send their children to the labour market, given a lack of income and difficulties in accessing school. This increase in child labour rates is also leading to an increase in the risk of accidents and mortality among children [8]. 370 million children have missed school meals during the pandemic, and have no access to hand-washing facilities to prevent the spread of COVID-19. Globally, 160 million children currently work, and this figure will increase by 9 million by the end of 2022 [9]. In addition to children, other groups affected by poverty include women, youth, unprotected workers, the underemployed, people with health problems, immigrants, people with lower-middle incomes, and people with low levels of education. These factors reflect an increase in the structural and social divide and worsening levels of inequality [8].

There is a direct relationship between poverty and inequality [10,11]. The latter has increased in recent years [12] because of the effects of the current pandemic and climate change [13]. Social vulnerability is closely related to the climate threat, and the stance of governments is critical to curbing inequalities [14]. The impacts of climate change will be most severe and immediate for billions of poor people, particularly those whose livelihoods are based on agriculture and subsistence activities, and who are directly dependent on weather patterns [15,16]. Women, youth, the elderly, ethnic and racial minorities, and indigenous and rural populations in underdeveloped and developing countries, are most affected by these factors [17].

Solutions are urgently needed to mitigate climate change, end poverty, and reduce inequality. This may also correct the adverse effects of the current pandemic and improve global economic growth, as a challenging element of the SDGs. The pandemic represents a challenge for the achievement of the SDGs at the global level. The circular economy and the new consumption model are among some of the measures that have been taken to address the SDGs [8,9]. Against this backdrop, the circular economy may contribute a solution to this problem. The circular economy concept proposes a paradigm shift from the current linear economy model based on non-renewable resources, to prioritise sustainability and resource renewal [18]. Based on three pillars (reduce, reuse, and recycle), this model makes it possible to continue conducting business in a manner that supports society's economic growth and environmental and social sustainability [19], in addition to reducing negative externalities and curbing social inequality [18].

Against this background, this study aimed to present the current state of scientific research related to poverty, inequality, and climate change, and to propose lines of improvement that can contribute to achieving three of the 17 SDGs (end poverty, SDG 1; reduce inequality, SDG 10; and climate action, SDG 13), proposed in the 2030 Agenda.

## 2. Materials and Methods

We carried out a systematic literature review (SLR) to conduct this study. This method was chosen because it synthesises the available evidence that summarises the information on a particular topic [20]. It is characterised by being systematic, critical, rigorous, specific, and reproducible [21,22]. We followed different phases to carry out the review [21,23], as detailed below:

- Phase 1: Identification of the research questions

In this phase, we posed these research questions (RQ) to achieve the objectives:

RQ1: What are the main papers that study poverty, climate change, and social inequality?

RQ2: What are the most relevant journals in which papers related to poverty, climate change, and inequality are published? [21–23]

RQ3: What has been the evolution of these papers during the period 1999–2020? [21–23]

RQ4: Which countries, universities, and areas of knowledge show the most significant interest in the field of poverty, inequality, and climate change? [21–23]

RQ5: Who are the most productive authors on the subject of poverty, inequality, and climate change? [21–23]

RQ6: What research topics are being addressed in the field of poverty, climate change, and social inequality? [21–23]

RQ7: Which groups are most affected by poverty, inequality, and climate change, according to the articles analysed?

- Phase 2: Search strategy

To carry out this stage, we considered the search terms of the information to be addressed [24]. For this purpose, we used the main concepts referred to in this research, recognising the different forms of writing, and synonyms and abbreviations [21]. In addition, the quality standards of the Prisma statement (Preferred Reporting Items for Systematic Reviews and Meta-Analyses) were used to include the relevant items and ensure the internal consistency of the systematic review [25].

We chose two of the most relevant academic databases for the information search process: Web of Science (WoS) and Scopus. These databases mainly collect scientific research products because of their multidisciplinary and international nature. Both are identified with an impact factor in their publications: the Journal Citation Report (JCR) for WoS and the SCImago Journal Rank (SJR) for Scopus [26].

The algorithm used in both databases was "poverty", "climate change", "inequality" with AND as the Boolean operator. The study period was from 1999, when the first publication appeared according to the search parameters used, until 2020.

As a result of the search, we obtained 720 papers, which were filtered according to the inclusion and exclusion criteria (Table 1). For this purpose, we retrieved scientific articles with a publication date between 1999 and 2020 [27], published in English. To meet the quality criteria of the literature, we searched for articles published in scientific journals [28], and those referring to primary studies [29], obtaining 480 contributions (279 from WoS and 201 from Scopus). We exported these documents to the Endnote database to continue applying the inclusion and exclusion criteria [30], as indicated in Table 1. In selecting research papers, we did not consider other types of publications such as books, book chapters, doctoral theses, colloquia, seminars, workshops, or conventions in this systematic review of the literature. We only considered articles published in scientific journals to ensure their quality [21,23].

**Table 1.** Inclusion and exclusion criteria.

| Inclusion | Exclusion |
|---|---|
| Period: 1999–2020 | Duplicates |
| Language: English | Not related to the topic "Poverty, Inequality and Climate Change" [23] |
| Primary Works [29] | Restricted access to full text [31] |
| Articles in scientific journals [28] | Unquoted articles [32] |

Following the definition of the inclusion and exclusion criteria, the next step was the selection of articles. For this, we followed several stages (Figure 1).

As shown in Figure 1, in the first stage, we identified 480 articles, which were exported to Endnote to eliminate duplicates ($n = 119$). In the second stage, we excluded all those articles that did not have restricted access to full text ($n = 46$) [31], and those that did not have any citations ($n = 89$). This is a determining aspect of the papers' relevance as a significant indicator of influence [32]. In the third and final stage, we manually analysed the title, abstract, and keywords of the 226 resulting articles. We discarded all research whose analysis did not focus on poverty, climate change, and inequality [23] ($n = 25$). We also carried out a complete reading of the articles, focusing mainly on their conclusions to meet the objectives pursued in this paper.

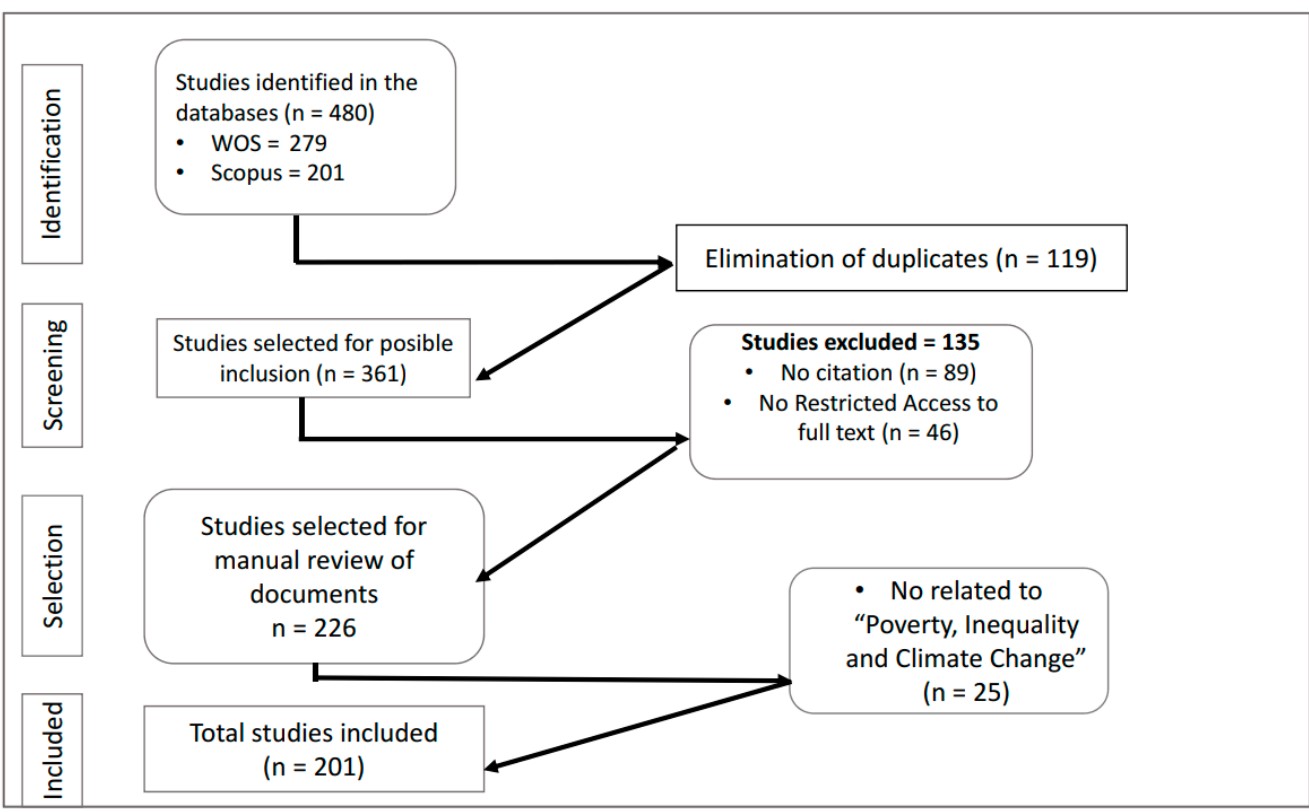

**Figure 1.** Flowchart of study selection process based on the PRISMA statement.

From this process, we obtained 201 articles for review, and detailed information that corresponds to the questions and objectives set out in this research.

### 3. Results

We obtained the main results of this work from the questions posed in the method-ological section and from the detailed reading of the 201 articles analysed. For this purpose, we prepared an Excel file in which we included the following items: title of the work, author, name of the journal, abstract, year of publication, number of citations, area of knowledge of the first author, country of origin of the first author, university to which the first author belongs, methodology, results of the works analysed, future lines of research, and principal conclusions.

These data allowed us to obtain the results presented here, which take into account the following parameters: year, scientific journal, country and university, area of knowledge, methodology, author and number of citations, the result of previous research, and the main limitations related to the subject studied.

### 3.1. Number of Publications per Year

As shown in Figure 2, the number of publications related to poverty, climate change, and social inequality began to increase from 2011 onwards, with 2018 being the year with the highest scientific output. Among the reasons that aroused academic interest in the topic in question were the concerns of the UN committee of scientists about the increase in extreme phenomena—rising sea levels, the decline in the ice in the Arctic, the increase in the global warming of the planet by 1 °C due to the emission of greenhouse gases, and the increase in inequality and poverty at a global level.

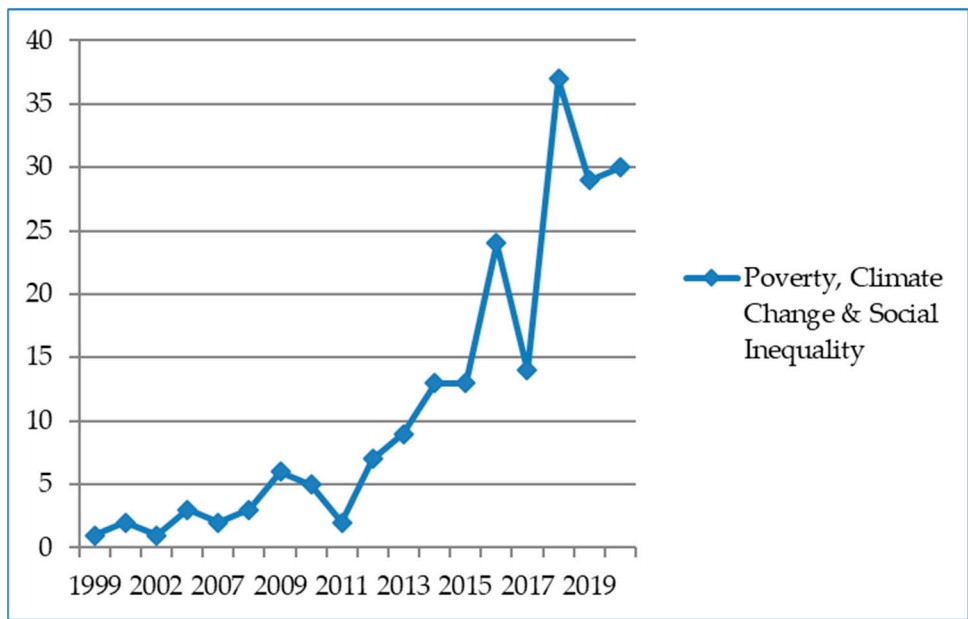

**Figure 2.** Number of publications per year on poverty, climate change, and social inequality.

### 3.2. Publications by Scientific Journals

Table 2 shows the breakdown of publications by journal, referring to those that have published more than two papers on the subject studied during the period 1999–2020, and their percentage of the total number of 201 definitive papers.

**Table 2.** Contributions per journal.

| Name of the Journal | No. of Publications | % |
|---|---|---|
| Sustainability | 8 | 3.98 |
| World Development | 7 | 3.48 |
| Climatic Change | 5 | 2.48 |
| Global Environmental Change-Human and Policy Dimensions | 5 | 2.48 |
| Environmental Science & Policy | 4 | 1.99 |
| Science of the Total Environment | 4 | 1.99 |
| Climate Policy | 3 | 1.49 |
| GeoJournal | 3 | 1.49 |
| Habitat International | 3 | 1.49 |

In addition to the information shown in Table 2, this study shows no specialisation or high concentration in one or several journals, because the maximum number of publications in one journal was eight during the period analysed. We should also note that 19 journals have 38 articles (two publications in each), and 121 journals only published one paper on the subject analysed.

### 3.3. Publications by Country and University

Table 3 lists the countries that contributed 10 or more publications on the topic of poverty, inequality, and climate change, and the universities that published two or more articles on this subject.

**Table 3.** Publications by country and university.

| Countries | No. of Publications | % | University |
|---|---|---|---|
| UK | 50 | 24.87 | University of London [6]<br>University of Leeds [5]<br>University of Southampton [4]<br>University of Sussex [4]<br>University of Harvard [4]<br>University of Oxford [3]<br>University of Western [2]<br>University of Lancaster [2]<br>University of Cardiff [2]<br>University of East Anglia [2]<br>University of the West of England [2]<br>University of Dundee [2] |
| USA | 36 | 17.91 | University of Washington [7]<br>University of California [3]<br>University of Maryland [3]<br>University of Cornell [2]<br>University of Michigan [2]<br>University of New York [2] |
| Australia | 17 | 8.45 | University of Canberra [3]<br>Australian National University [2]<br>University of Queensland [2]<br>Charles Sturt University [2] |
| Germany | 12 | 5.97 | University of Berlin [3]<br>University of Bonn [2]<br>Technische Universität Dresden [2] |

A share of 89.97% of the countries contributed fewer than 10 publications, with Norway, Italy, Sweden, India, and China being the most productive within this ranking (with more than five publications and fewer than 10), and Argentina, Colombia, France, Kenya, Russia, Switzerland, Ghana, Czech Republic, Mexico, Malaysia, Nigeria, Pakistan, Egypt, and the Philippines, being the least productive, with only one publication.

A share of 10.03% of the countries contributed more than 10 publications. The United Kingdom and the USA were most significantly different from the remaining countries in Table 3.

*3.4. Publications by Field of Knowledge*

To select the different areas of knowledge or departments that comprise this work, we considered the area of knowledge or department to which the first author belongs. If this information was not available, we used the second, etc. We used this criterion because it is the same as the criterion we followed for selecting universities [21]. We aimed to maintain homogeneity in the data analysis of the entire work.

Table 4 shows the most prolific areas of knowledge. We identified 14 different areas or departments addressing climate change, poverty, and social inequality. It is worth noting that the areas of Sustainability and Economics have the highest number of publications (51.7%), with the remainder distributed among the 12 other fields of knowledge. The significant interest shown in these areas is mainly due to the knowledge of the effects of climate change on the most vulnerable populations, mainly in underdeveloped countries such as India or South Africa, where the main livelihood is agriculture. Heavy rains, hurricanes, and droughts make it increasingly difficult for the rural poor to eat, which is a relevant factor for food security, and results in a higher level of inequality [33,34].

**Table 4.** Publications by field of knowledge.

| Field of Knowledge | No. of Publications | % |
|---|---|---|
| Sustainability and Environment | 60 | 29.85 |
| Economics | 44 | 21.89 |
| Geography | 27 | 13.43 |
| Health and Medicine | 25 | 12.44 |
| Agriculture | 8 | 3.98 |
| Development and Planning | 8 | 3.98 |
| Sociology and Anthropology | 8 | 3.98 |
| Psychology | 7 | 3.48 |
| Engineering | 5 | 2.49 |
| Biology | 3 | 1.49 |
| Education | 3 | 1.49 |
| Law | 2 | 1.00 |
| Political science | 1 | 0.5 |

### 3.5. Type of Study

Academic research divides work into two main areas. One of these is empirical research studies (related to evidence from questionnaires, experiments, or case studies). The other is documentary research (which mainly includes contributions related to literature reviews and conceptual documentation) [35].

Thus, of the 201 articles analysed, 60.19% were classified as empirical studies and 39.81% as documentary studies (Figure 3).

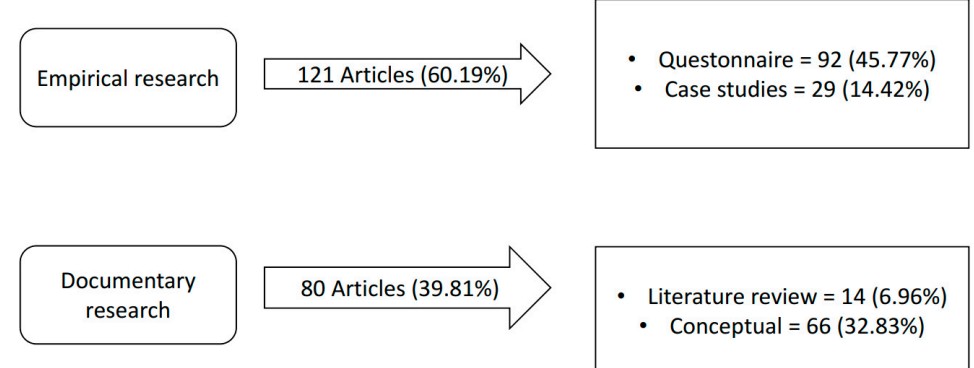

**Figure 3.** Type of study.

As shown in Figure 3, the questionnaire was the study technique chosen by the majority of authors [36]. These questionnaires were used in numerous scientific experiments, as the analysis of the 201 articles shows, whether in the form of an interview or a survey. Focus groups were the most widely used qualitative research technique, and various statistical models of regression, multivariate, polarisation, clustering, and multisectoral analysis, among others, were used to process the resulting data [37].

It should also be noted that the case studies focused on developing or underdeveloped countries, such as Mozambique, Kenya, Rwanda, and Bangladesh, where the populations are characterised by high levels of poverty and inequality, and are thus more vulnerable to the adversities due to climate change.

### 3.6. Publications by Author and Relevance of Articles

This section lists the authors who are actively involved in research on poverty, cli-mate change, and inequality. A total of 528 authors participated in 201 articles in this systematic review during the period analysed, and seven of these alluded to the circular economy. The most productive researcher in the subject analysed was A. Jerneck, who participated in

four papers, followed by J.A. Silva, with three publications. Of the remaining re-searchers, 28 contributed two articles and 498 contributed one paper, representing 94.31% of the total.

Classifying the articles by their relevance, according to the number of citations re-ceived [32], a total of 6008 citations were obtained from the 201 articles retrieved in this systematic review of the literature, of which 61.19% (123 papers) received between one and 10 citations, and 41 papers were cited between 11 and 20 times, representing 20.39%. A total of 37 articles had more than 21 citations, accounting for 18.94%.

Ten articles received more than 100 citations; two of these were published in the journal *Climate Change* (Table 5). Adger was the most cited author, receiving 1340 citations for his two published papers.

**Table 5.** Most cited publications.

| Authors | Year | Title | Journal | No. of Citations |
|---|---|---|---|---|
| Kelly, P. M.; Adger, W. N. | 2000 | Theory and practice in assessing vulnerability to climate change and facilitating adaptation | *Climatic Change* | 775 |
| Adger, W. N | 1999 | Social vulnerability to climate change and extremes in coastal Vietnam | *World Development* | 565 |
| Arora-Jonsson, S. | 2011 | Virtue and vulnerability: Discourses on women, gender and climate change | *Global Environmental Change-Human and Policy Dimensions* | 296 |
| McMichael, A. | 2000 | The urban environment and health in a world of increasing globalization: issues for developing countries | *Bulletin of the World Health Organization* | 257 |
| Denton, F. | 2002 | Climate change vulnerability, impacts, and adaptation: Why does gender matter? | *Climatic change* | 253 |
| McMichael, A.; Friel, S.; Nyong, A.; Corvalan, C. | 2008 | Global environmental change and health: Impacts, inequalities, and the health sector | *Bmj* | 182 |
| Chakravarty, S.; Chikkatur, A.; de Coninck, H.; Pacala, S.; Socolow, R.; Tavoni, M. | 2009 | Sharing global CO2 emission reductions among one billion high emitters | *Proceedings of the National Academy of Sciences of the United States of America* | 175 |
| Gentle, P.; Maraseni, T. | 2012 | Climate change, poverty and livelihoods: adaptation practices by rural mountain communities in Nepal | *Environmental Science & Policy* | 166 |
| Hirsch, P.; Adams, W.; Brosius, J.; Zia, A.; Bariola, N.; Dammert, J. | 2011 | Acknowledging Conservation Trade-Offs and Embracing Complexity | *Conservation Biology* | 154 |
| Buhaug, H.; Urdal, H.Conservation Biology | 2013 | An urbanization bomb? Population growth and social disorder in cities | *Global Environmental Change-Human and Policy Dimensions* | 144 |

### 3.7. Main Contributions

In addition to this systematic review of the literature, which highlights the current state of the research in this area, the main contributions of this work are the identification of the groups most affected by climate change, poverty, and inequality, and the sectors most affected. We also show evidence of the relationship between these concepts in the articles analysed. The findings also identify the main papers that address COVID-19 and the SDGs.

Regarding the groups most affected by poverty, inequality, and climate change, it should be noted that 53.73% of the works studied do not show any evidence of this. Of

the remainder, 22.55% of the contributions highlight women as the main group affected by this situation. Women are the most vulnerable to climate change, particularly those with a low cultural, educational, and socio-economic level [38]. In addition, they have difficulty adapting to working the land due to gender inequality in the affected countries [39], and also show less resilience to the effects of climate change. African women are more vulnerable to flooding, especially those with inadequate housing and low incomes [40].

Regarding the groups most affected by unequal poverty, 12.61% of the studies analysed identify sick people as the second most vulnerable group affected by the problems under study. The most common illnesses include malaria, asthma, hepatitis A, and cholera [41–43]. The remainder of the group (i.e., 11.11%) are children who are also affected by climatic adversities. They usually belong to impoverished families whose income relies on agriculture. Climate-induced changes cause them to suffer from low food intake and malnutrition. This is a severe problem, particularly given that another of the goals of the 2030 Agenda, specifically SDG2, is to end world hunger [44,45].

In addition to these groups, other groups affected by this situation are indigenous people [39], the elderly, immigrants, people from ethnic or racial minorities [17], those who have problems with water supply [40], and those living in slums [43].

Concerning the sectors most affected by the problems analysed, 76.62% of the works analysed do not provide clear evidence of these sectors. However, in 21.89%, agriculture is shown to be the sector most affected by climate disasters, and 1.49% of papers identify fishing, livestock, and tourism.

Regarding the second part of this section, and to analyse the relationship between the problems outlined in this work, we examined the objectives, results, and conclusions of the 201 articles to determine the percentage of publications studying this link. For this purpose, we started from the premise that climate change worsens the situation of poverty and inequality of the most vulnerable people [40,41], and that people suffering from poverty and inequality have a greater detrimental effect on environmental pollution because the means at their disposal for their subsistence cause them to expel higher levels of $CO_2$ into the atmosphere [45].

We can deduce from this descriptive analysis that 73.03% of the works analyse this aspect. With this we can confirm, in addition to the proposals and contributions of the reviewed authors on this subject, that there is a close relationship between poverty, climate change, and inequality.

In addition to the results set out in this section, we also considered the situation of the current pandemic and the relevance of meeting the goals of the 2030 Agenda. For this reason, we wanted to determine the percentage of publications that refer to this issue.

Concerning the number of papers referring to COVID-19, of the 30 articles published in 2020, only four (13.33%) mention COVID-19. Furthermore, 83.33% of the papers were published in the period of May–December of this year, which is considered a low number of contributions in this area.

Only 14.42% of the 201 articles reviewed consider the contributions to the SDGs, whereas 85.58% do not mention them. The largest number of publications was concentrated in the period 2019–2020. Nevertheless, there is a large gap in this literature, and there is a need to propose lines of future research that can address this issue, given its importance for achieving the SDGs.

## 4. Discussion

In this section, we pose two fundamental questions for this research. The first answers the question of whether there is a relationship between inequality and climate change. The second examines the role of the circular economy as an element that can contribute to reducing inequality and the effects of climate change, and thus contribute to the achievement of the three SDGs set out in this work (end poverty, SDG 1; reduce inequality, SDG 10; and climate action, SDG 13).



To resolve the first question, and based on the results obtained in this research, we can affirm a close and direct relationship between the three SDGs analysed, because 73.03% of the works analysed demonstrate this. Furthermore, we can see that the effects of climate change generate patterns of inequality in health, gender, and income, as indicated below.

This work shows that health has a basic role in the face of climate change, poverty, and inequality, with 81.59% of the works analysed related to this. Among the main problems related to the health of the most vulnerable people affected by this problem, those related to contaminated water, food insecurity, malnutrition, and hunger can be highlighted [46]. Due to the lack of water infrastructure, this population has to drink water from rivers contaminated with microorganisms and chemicals, causing severe diseases such as cholera, typhoid fever, and hepatitis A [39,40]. In addition, climatic adversities impact harvests and prevent sufficient access to healthy and wholesome food, cause food prices to rise, and restrict consumption, as experienced by indigenous people in Canada and rice farmers in the Philippines [41,47,48]. Aboriginal peoples living in remote areas face greater challenges and are more vulnerable to the climate than urban populations, due to their close ties to the land, and more pronounced socio-economic and political marginalisation [49,50]. Lack of provision of transportation, whether public or private, and road infrastructure hinders access to early medical care to prevent and cure diseases [47,51].

Concerning gender inequality, we highlight in this SLR that the role of women in climate change mitigation policies is not taken into account, because they are one of the most affected groups [44]. A detailed reading of the 201 articles analysed shows that, since the 1970s, women have not benefited equally from agricultural development programmes in underdeveloped countries [52]. This discrimination was due to the attachment of women to the roles of mother and wife [53], and to the existence of a global gender gap between vulnerabilities and access to resources. This led to a gender-differentiated capacity for climate adaptation, in which women and girls have been placed at a disadvantage [46]. Moreover, because of this role, which is mainly associated with the performance of domestic tasks, women face increasing risks to their health [36]. Cooking over an open fire with solid fuels results in incomplete combustion, causing severe respiratory diseases that exacerbate mortality among women and children in rural areas of sub-Saharan Africa, Nigeria, Ethiopia, Malaysia, Mozambique, Bangladesh, and Kenya, among others [14,54–57].

Concerning income inequality, the two-way relationship between poverty and climate change [58] is presented in the articles analysed. The articles highlight that the effects of climate change (droughts, floods, hurricanes, etc.) will be more challenging in the most impoverished countries, and in those where economies are based on the primary sector and dependent on agriculture as the main activity. This is the case of countries located in the southern tropics, such as sub-Saharan Africa, Kenya, Ethiopia, and Bangladesh, and among low-income people, i.e., the poorest [59,60]. Furthermore, growth in poverty and income inequality increases $CO_2$ emissions in these countries [45]. Insufficient economic resources highlight the constraints and capabilities of household resources to reduce $CO_2$ levels, such as the use of renewable wood cookers that improve the efficiency of charcoal consumption [36]. It should also be noted that poor women, particularly those without access to adequate health care and decent housing, are more vulnerable to flooding and are slower to recover than other women with higher incomes or those of higher social classes [44].

To answer the second question, we rely on the principles of circular economy. These principles are based on three premises. The first is that the circular economy preserves and conserves natural capital because it efficiently selects resources and uses renewable resources. The second optimises resource yields by distributing products, components, and materials with their maximum utility at all times in both technical and biological cycles. The third is based on eliminating negative externalities [61]. This system may benefit populations, in particular, who are the most vulnerable and impoverished, and depend on agriculture for their livelihoods. In the current work, we show that this group is among the most adversely affected by the impacts of climate change [62]. The introduction of a

production system based on natural and renewable resources, waste decomposition, and the use of more efficient technology would increase production and employment, and reduce poverty and inequality [61,63]. Therefore, better soil care can contribute to greater equity and reduce CO2 emissions [14].

Similarly, applying this circular economy model may eliminate negative externalities, and reduce negative impacts on food, mobility, education, health, and leisure. It can also be used to manage external factors such as land use, pollution of air and water, noise pollution, and the dumping of toxic substances. Controlling these externalities may curb inequality because the different socio-economic impacts reflect a matrix of population inequality, whose structural axes are related to income, gender, health, race, education, migration, and territory [39]. Inequality, social sustainability, vulnerability, and the use of natural resources are closely linked to the effects of climate change in areas such as Asia, sub-Saharan Africa, Mozambique, and Ghana [43,64,65]. This is the case of the farmers who occupy these territories, who should adapt to market needs by introducing technical improvements and natural resources that do not damage the ecosystem. Thus, farmers would not remain on the margins of society, and patterns of social vulnerability would not be generated [39,66]. Patterns of vulnerability are created by unequal opportunities and differentiation, both socially between households and geographically between villages [39].

As a result of the current work, and considering the foundations on which the principles of the circular economy are based, we believe that this economic model constitutes an element that can contribute to reducing inequalities and mitigating the effects of climate change, and thus achieve the SDGs, as proposed in the 2030 Agenda. In addition to the two issues raised in this discussion, this paper highlights the urgent need to open up future lines of research that address the issue of COVID-19 and the 17 SDGs within the context of the aspects analysed, such as poverty, climate change, and inequality. The current pandemic highlights the lack of protection of social systems, particularly those related to health [67], and hampers the fulfilment of SDG3—health and well-being—and increases the levels of inequality and poverty at the global level. Moreover, the pandemic is worsening issues related to global hunger, education, gender equality, access to clean water, access to clean energy, and the right to decent work, and undermines economic growth. These factors are all proposed in the 2030 Agenda. The need for new contributions in this direction may contribute to addressing this problem. This research contributes to the debate surrounding these issues.

## 5. Conclusions

In this systematic review of the literature, 201 publications were identified concerning the topic of poverty, inequality, and climate change. As expected, the largest number of publications was concentrated in the UK and the USA, with the remainder of the countries positioned in second place. Furthermore, the areas related to the economy and sustainability were discussed in the highest number of scientific publications (51.7%). This may be because most of the analysed works took into account the issue of social inequalities in the face of climate change threats. However, the policies aimed at mitigating CO2 emissions do not quantify the direct and indirect impacts on the most vulnerable populations [60], such as women (gender inequality), the poorest (income inequality), and the sick (health inequality), as reflected in this document. The effects of climate change have stark consequences for people's health, particularly the health of the most vulnerable [58]. Analysing the interactions between climate change and health should feature prominently in studies related to environmental epidemiology and public health [67].

Another topic of significant interest in this SLR is COVID-19. This pandemic shows that countries are not immune to the threat of infectious diseases, regardless of their level of wealth [68]. However, low- and middle-income countries may be the most vulnerable to COVID-19. Because the population density in these countries is higher, a greater proportion of the population suffers from health conditions, and health systems are unprepared for the pandemic [69]. Difficulty accessing the public health system leads to higher costs and

shortages of skilled labour and medical supplies. There is a global gap in health systems, leading to increased social unprotection [9].

However, despite the importance of COVID-19, we found few studies that provide evidence of its effects in the current review. We believe that one of the main reasons for this is the lack of data for this period. Another factor is the time lag between the reception and publication of the studies.

The scarce contribution to the SDGs in the analysed works should also be highlighted; only 14.42% of the reviewed publications addressed this issue. We consider that, within the field of climate change, poverty, and inequality, priority should be given to the issue of the SDGs, given their importance.

In addition to the shortcomings mentioned above, it should be noted that issues related to energy, poverty, and food insecurity are rarely addressed in the articles studied in this review. Regarding energy, only 10.95% of the publications pay attention to the phenomenon of energy poverty, which is experienced in urban settings [68] of European countries [69]. Regarding poverty, 47.26% of the analysed works refer to food insecurity. This is inadequate given the increase in global hunger and the difficulty of obtaining food due to the effects of climate change. Thus, it would be advisable to carry out further research into this respect.

In summary, further research should be conducted into the interconnection between climate change, poverty, and inequality, and to consider the role of the circular economy as an element that can contribute to ending poverty (SDG 1), reducing inequality (SDG 10), and enhancing climate action (SDG 13). For this purpose, in addition to considering the circular system as a possible solution, climate action policies that take into account the socio-economic aspects of the population should also be considered [70]. Inclusive measures and policies that collectively address these three objectives may be more beneficial to achieving the sustainable development goals [71]. Government corruption should also be assessed, because it is increasingly present in developing countries and prevents appropriate management from achieving these goals [72].

**Author Contributions:** Conceptualization, M.d.C.P.-P.; methodology, M.d.C.P.-P.; software, J.R.-C.; validation, M.J.-G.; formal analysis, M.d.C.P.-P.; investigation, M.d.C.P.-P.; resources, M.d.C.P.-P. and A.R.P.-S.; data curation, J.R.-C. and A.R.P.-S.; writing—original draft preparation, J.R.-C.; writing—review and editing, M.J.-G.; visualization, M.J.-G.; supervision, M.J.-G.; project administration, J.R.-C.; funding acquisition, A.R.P.-S. All authors have read and agreed to the published version of the manuscript.

**Funding:** This publication and research were partially funded by INDESS (the University Institute of Research for Social Sustainable Development) of the University of Cádiz, Spain; and, also by the Plan for the Promotion and Promotion of Research and Transfer of the University of Cádiz (Programa de Fomento e Impulso de la Actividad de Investigación y Transferencia de la Universidad de Cádiz).

**Institutional Review Board Statement:** Not applicable.

**Informed Consent Statement:** Not applicable.

**Data Availability Statement:** Not applicable.

**Conflicts of Interest:** The authors declare no conflict of interest.

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
