# Peer review of "Analysis of Research on the SDGs: The Relationship between Climate Change, Poverty and Inequality"

_applsci, doi:10.3390/app11198947_

Round 1

Reviewer 1 Report

The article evaluates the peer-reviewed publications on poverty, inequality, and climate change and potential implications for attaining SDGs 1, 10, and 13. Although the article has raised relevant issues related to poverty, inequality, and climate change, it needs improvements to move forward in the publication process.

The weakest aspect of this paper is methodology. The themes to evaluate poverty, inequality, and climate change across 201 publications are unclear. How are the findings and conclusions drawn? It is also not clear how the content analysis is administered among these publications.

More empirical research about SDGs, poverty, and inequality is published in gray literature. I am not sure why the authors did not consider them in the course of the analysis. Some explanations are required.

In the introduction, the discussion on the role of COVID-19 to worsen poverty and inequalities has to be added. Similarly, there is no mention of the COVID-19 impacts in the rest of the analysis. The mention of COVID-19 impacts in the abstract is misleading without further comments, discussion, and conclusions.

Why authors chose only three SDGs out of 17? There should be more discussion.

Under Materials and Methods, the research questions for 'Phase I' can be consolidated. They are too generic and vague. For example, "Who are the most productive author?" It is also not clear what are the driving assumptions of these research questions.

The flow chart (Figure 1) should be reorganized boxes to show their relationships. It is not clear how a step leads to the other.

The explanation of Table 2 is confusing. Table 2 does not contain 18 journals and 36 articles. How do we know that there were 123 journals with only one article published from Table 1? There are inconsistencies; if there is a % sign on the column heading, it is unnecessary to include it in the cell values.

In Table 3, why Australia and Germany are mentioned twice? What is the rationale for including the name of universities? The description of Table 3 is incorrect. There is no mention of Thailand, the Netherlands in the table. Please revise and correct.

In Table 4, how were the fields identified? What are the criteria and bases? There is a detachment between the table and the methodology. The authors should explain the methods to identify fields of knowledge presented in Table 4. As it is, Table 4 seems to be drawn based on the publication titles. The same comments are relevant to Figure 3.

It is unclear why writers have analyzed the article's authors as indicators to evaluate the publications on poverty, inequality, and climate change (Section 3.6). Why is it relevant, and what are the broader contributions to the research questions?

There are inconsistencies and grammatical errors. Proofreading and editing are required. For example, the name of a journal should be in the title case, such as Climate change.

There is a disconnect between the discussion of findings and the analysis of the publications. How the findings in discussion sections are related to analyses. Are there themes across these 201 publications showing the disproportionate impacts of climate change, poverty, and inequality on specific groups and certain sectors? If that is the case, authors should clearly show in the method and analysis sections while presenting the contents from these publications. It is also unclear how themes are drawn. It is evident that poverty and climate change are related. But what are the nuance offerings (codes, figures, and excerpts consolidating themes from 201 publications) from this review? What are the new contributions? Why is the timeline of 1999 to 2020 chosen? Why is COVID-19 literature not included?

The discussion section reads like an extended literature review than an organized discussion of the findings from the review of 201 publications.

The discussion of circular economy is opined by authors rather than the findings from the review. Some of the explanation about the circular economy and agriculture does not make clear sense. For example, how the farmers' interactions with the market and commercialization make them vulnerable and how geographic and household differentiations make them vulnerable. It is unclear.

Author Response

Reviewer 1:

We want to thank the reviewer for his/her comments and suggestions about the work previously entitled "Analysis of research on the SDGs: The relationship between climate change, poverty and inequality". These modifications undoubtedly improve the quality and accuracy of the article.

We have highlighted all changes corresponding to such suggestions in blue colour fonts in our manuscript:

1.- The weakest aspect of this paper is methodology. The themes to evaluate poverty, inequality, and climate change across 201 publications are unclear. How are the findings and conclusions drawn? It is also not clear how the content analysis is administered among these publications.

We have included a paragraph in section 3 (lines 152-160) according to your indications. We have also deleted “We have analysed the articles obtained, taking into account the following parameters”.

2.- More empirical research about SDGs, poverty, and inequality is published in gray literature. I am not sure why the authors did not consider them in the course of the analysis. Some explanations are required.

Indeed, issues related to poverty and inequality are also published in the grey literature, as indicated by the reviewer. However, we chose not to include it in this paper, following the methodology of Martín-Navarro, et al. (2018) and Díaz-Iso et al. (2020), as indicated below:

Martín-Navarro, A.; Sancho, M. P. L.; & Medina-Garrido, J. A. BPMS para la gestión: una revisión sistemática de la literatura. REDC2018, 41(3), 213. https://doi.org/10.3989/redc.2018.3.1532 [CrossRef]

“Perhaps, if the search is extended to other types of publications (books, book chapters, contributions to conferences, doctoral theses or even divulgation works), we could find more contributions confirming the findings of this research or that, on the contrary, represent a new perspective for the literature in the field.  In any case, we chose to include only academic articles published in journals to guarantee their quality.” (Martín-Navarro et al., 2018)

Díaz-Iso, A.; Eizaguirre, A.; & García-Olalla, A. M. Una revisión sistemática del concepto de actividad extracurricular en Educación SuperiorEducación XX1 2020, 23(2). https://doi.org/10.5944/educxx1.25765 [CrossRef]

“All papers published at conferences, symposia, doctoral theses, seminars, colloquia, workshops or conventions were excluded.” (Díaz-Iso et al., 2020).

However, we have added the following paragraph (lines 130-134). In the selection of research papers "In selecting research papers, we have not considered other types of publications such as books, book chapters, doctoral theses, colloquia, seminars, workshops or conventions in this systematic review of the literature. We have only considered articles published in scientific journals to guarantee their quality [21, 23].”

3.- In the introduction, the discussion on the role of COVID-19 to worsen poverty and inequalities has to be added. Similarly, there is no mention of the COVID-19 impacts in the rest of the analysis. The mention of COVID-19 impacts in the abstract is misleading without further comments, discussion, and conclusions.

As suggested by the reviewer, we have added the impacts of covid-19 in the introduction (Lines 48-60 and 71-75), discussion (See comment number 13) and conclusions (Lines 418-432) of this paper. We have also included this term in the keywords.

4.- Why authors chose only three SDGs out of 17? There should be more discussion.

In 2015 the United Nations (UN) established the 2030 Agenda for Sustainable Development to eradicate extreme poverty, reduce inequality, and protect the planet (Martin et al., 2020).  The 2030 Agenda highlights the importance of biodiversity and ecosystem functioning for sustaining economic activities and the well-being of local communities.  On the other hand, there are indications that climate change is causing an increase in world hunger and inequality (Ridaura, 2020). Therefore, one of the decisions that prompted us to choose these three goals from among the 17 SDGs was mainly this one.  However, we must make it clear that the 17 goals proposed in the 2030 agenda are very important and interrelated for these researchers. Still, it was practically impossible in this work to cover all of them due to their length. However, we will keep this in mind for future lines of research, given the relevance of the topic in question. 

See discussion section.

5.-Under Materials and Methods, the research questions for ‘Phase I’ can be consolidated. They are too generic and vague. For example, “Who are the most productive author?” It is also not clear what are the driving assumptions of these research questions.

Considering the reviewer's comments, we have modified the research questions and added the authors' references. We have replaced these questions with the following (Lines 96-107):

RQ2: What are the most relevant journals where papers related to poverty, climate change and inequality are published? [21-23]

RQ3: What has been the evolution of these papers during the period 1999-2020? [21-23]

RQ4: Which countries, universities and areas of knowledge show the most significant interest in the field of poverty, inequality and climate change? [21-23]

RQ5: Who are the most productive authors on the subject of poverty, inequality and climate change? [21-23]

We have also added another question, RQ7: Which groups are most affected by poverty, inequality and climate change, according to the articles analysed?

6.-The flow chart (Figure 1) should be reorganized boxes to show their relationships. It is not clear how a step leads to the other.

We have changed figure 1, as suggested by the reviewer, to the following.

7.-The explanation of Table 2 is confusing. Table 2 does not contain 18 journals and 36 articles. How do we know that there were 123 journals with only one article published from Table 1? There are inconsistencies; if there is a % sign on the column heading, it is unnecessary to include it in the cell values.

As the reviewer rightly points out, the results are not from table 2, but from the research. We have also changed the numbers 123 to 121 and 18 to 19, as they were wrong.  With this comment, we want to reflect the existence of many journals with 1 and 2 articles.

The 121 journals to which we refer and which include only 1 paper are:  

1.      Academy of Management Perspectives

2.      Actualidad Juridica Ambiental

3.      African Journal on Conflict Resolution

4.      Agriculture Ecosystems & Environment

5.      Ain Shams Engineering Journal

6.      Ambio

7.      Appetite

8.      Applied Energy

9.      Applied Geography

10.   Asia Pacific Viewpoint

11.   Australian Journal of Management

12.   Bmj

13.   British Journal of Sociology

14.   Building and Environment

15.   Bulletin of Latin American Research

16.   Bulletin of the World Health Organization

17.   Ciriec-España

18.   Civitas

19.   Communications of the Acm

20.   Conflict Management and Peace Science

21.   Conservation & Society

22.   Conservation Biology

23.   Contemporary Economics

24.   Critical Perspectives on Accounting

25.   Critique of Anthropology

26.   Current Opinion in Environmental Sustainability

27.   Development Policy Review

28.   Disability and Society

29.   DLSU Business and Economics Review

30.   Earth System Dynamics

31.   Earths Future

32.   Ecological Economics

33.   Ecological Indicators

34.   Ecology and Society

35.   Ecosystem Services

36.   Elementa-Science of the Anthropocene

37.   Energy Economics

38.   Energy for Sustainable Development

39.   Energy, Ecology and Environment

40.   Engineering

41.   Environment and Planning A

42.   Environment and Urbanization ASIA

43.   Environment Development and Sustainability

44.   Environmental Development

45.   Environmental Health

46.   Environmental Justice

47.   Environmental Research Letters

48.   Environmental Science and Pollution Research

49.   Ethics and International Affairs

50.   European Urban and Regional Studies

51.   Forest Policy and Economics

52.   Frontiers in Astronomy and Space Sciences

53.   Frontiers in Psychology

54.   Frontiers in Sustainable Food Systems

55.   Geoforum

56.   Geographical Journal

57.   Geography Compass

58.   Geophysical Research Letters

59.   Global Environmental Change

60.   Global Food Security-Agriculture Policy Economics and Environment

61.   Global Journal of Emerging Market Economies

62.   Government Information Quarterly

63.   Health & Place

64.   Health Promotion International

65.   Human Ecology

66.   Indian Journal of Public Health Research and Development

67.   Interdisciplinaria

68.   International Journal of Climate Change Strategies and Management

69.   International Journal of Disaster Risk Reduction

70.   International Journal of Educational Development

71.   International Journal of Environment and Sustainable Development

73.   International Journal of Technology and Design Education

74.   International Organisations Research Journal

75.   Journal of Arid Environments

76.   Journal of Business Economics and Management

77.   Journal of Business Ethics

78.   Journal of Contextual Behavioral Science

79.   Journal of Developing Societies

80.   Journal of Energy in Southern Africa

81.   Journal of Human Development and Capabilities

82.   Journal of Hunger & Environmental Nutrition

83.   Journal of International Development

84.   Journal of Nutrition

85.   Journal of Rural Studies

86.   Journal of Teacher Education for Sustainability

87.   Land Use Policy

88.   Landscape and Urban Planning

89.   Latin American Perspectives

90.   Local Economy

91.   Magallania

92.   Management and Organization Review

93.   Mathematics

94.   Mediterranean Journal of Social Sciences

95.   Mountain Research and Development

96.   Natural Hazards

97.   Nature Climate Change

98.   Perspectives in Public Health

99.   Philosophical Transactions of the Royal Society a-Mathematical Physical and Engineering Sciences

100. Philosophical Transactions of the Royal Society B-Biological Sciences

101. Plos One

102. Proceedings of the National Academy of Sciences of the United States of America

103. Psychology of Global Mobility

104. Public Health

105. Renewable Agriculture and Food Systems

106. Reproductive Health Matters

107. Review of Business

108. Revista Espanola De Investigaciones Sociologicas

109. Revista Peruana de Medicina Experimental y Salud Publica

110. Samj South African Medical Journal

111. Social Science & Medicine

112. Social Science Quarterly

113. Sustainable Cities and Society

114. Sustainable Development

115. Technological Forecasting and Social Change

116. The BMJ

117. Water Alternatives-an Interdisciplinary Journal on Water Politics and Development

118. Weather Climate and Society

119. Wiley Interdisciplinary Reviews-Climate Change

120. World Bank Economic Review

121. World Review of Nutrition and Dietetics

The 19 journals that include 2 publications are as follows:

1.      Australian and New Zealand Journal of Public Health

2.      Analyse und Kritik

3.      Bioethics

4.      Cities

5.      Climate and Development

6.      Energy and Buildings

7.      Energy Policy

8.      Energy Research & Social Science

9.      Environment, Development and Sustainability

10.   Environmental & Resource Economics

11.   International Environmental Agreements-Politics Law and Economics

12.   Journal of Cleaner Production

13.   Journal of Environment & Development

14.   Journal of Environmental Management

15.   Land

16.   Local Environment

17.   Regional Environmental Change

18.   Sage Open

19.   Social Indicators Research

We have also replaced the following paragraph from lines 178-182 following the reviewer's comments: “We can see in Table 2 that 18 journals concentrate 36 articles (2 per journal), and in 123 journals only 1 article has been published in each of them. This shows no specialisation or high concentration in one or several journals, as the maximum number of publications in one journal was 8 over the period analysed”.

With: “In addition to the information shown in table 2, this study shows no specialisation or high concentration in one or several journals, as the maximum number of publications in one journal was 8 over the period analysed. We should also note that 19 journals have 38 articles (2 publications in each), and 121 journals have only obtained 1 publication on the subject analysed.”.  (Lines 178-182).

On the other hand, we have removed the % from the 3rd column of table 2.  

8.- In Table 3, why Australia and Germany are mentioned twice? What is the rationale for including the name of universities? The description of Table 3 is incorrect. There is no mention of Thailand, the Netherlands in the table. Please revise and correct.

Thanks to the reviewer's comments, we have corrected the error and have left Germany and Australia only once.

The reason for including the names of the universities in this table is to provide greater simplicity and clarity to the work. Instead of presenting 2 separate tables (one for countries and the other for universities), we have chosen to show them together, so as not to repeat the names of the countries in the table that mentions the universities.

As indicated by the reviewer, we have removed Thailand and the Netherlands from line 196, as they were not correct. 

9.- In Table 4, how were the fields identified? What are the criteria and bases? There is a detachment between the table and the methodology. The authors should explain the methods to identify fields of knowledge presented in Table 4. As it is, Table 4 seems to be drawn based on the publication titles. The same comments are relevant to Figure 3.

We have identified the fields taking into account the authors' departmental information. These are indicated in the journals of the articles analysed, in author information.  The criterion we followed was the department to which the first author belonged, as in the selection of universities. We have opted for this to maintain homogeneity in the data analysis.

We have added to the text in section 3.4. area of knowledge: “To select the different areas of knowledge or departments that make up this work, we have considered the area of knowledge or department to which the first author belongs. If this information was not available, we have used the second and so on. We have opted for this criterion because it is the same as the one we followed for selecting universities [21]. We want to maintain the same homogeneity in the data analysis of the entire work.” (lines 198-203)

Concerning the assignment of the category "area of knowledge", Boudon's definition was used as a starting point. He considers that the contents of an area of knowledge are obtained from the populations of researchers in the area in question.

On the other hand, and considering the reviewer's appreciation, we have opted to change some of the names of the different areas of knowledge in table 4, namely: Sustainability= Sustainability and Environment, Health= Health and Medicine, and grouping Sociology and Anthropology.

About the comments made about figure 3, we must point out that we do not understand the changes proposed: For this reason, we have not made any changes to it. We await any clarification from you to be able to do so.    

10.-It is unclear why writers have analyzed the article’s authors as indicators to evaluate the publications on poverty, inequality, and climate change (Section 3.6). Why is it relevant, and what are the broader contributions to the research questions?

Concerning the question raised by the reviewer, we have to point out 2 aspects:

  1. We have carried out the authors' analysis because it is one of the classic analyses of the methodology in this type of research. It is part of the data extraction of the works analysed, as seen in the contributions made by Martín-Navarro (2018) and Manterola et al. (2013), authors referenced in this paper.
  2. It is interesting to know whether there are one or more people who stand out above the average in the field of research analysed, who we can call experts in the field both in terms of the quality and quantity of their published work. In this way, we can answer RQ5, also to be able to determine 10 leading authors in this field.

11.- There are inconsistencies and grammatical errors. Proofreading and editing are required. For example, the name of a journal should be in the title case, such as Climate change

A translator has further revised the text to improve grammar and editing.

12.-There is a disconnect between the discussion of findings and the analysis of the publications. How the findings in discussion sections are related to analyses.  Are there themes across these 201 publications showing the disproportionate impacts of climate change, poverty, and inequality on specific groups and certain sectors? If that is the case, authors should clearly show in the method and analysis sections while presenting the contents from these publications. It is also unclear how themes are drawn. It is evident that poverty and climate change are related. But what are the nuance offerings (codes, figures, and excerpts consolidating themes from 201 publications) from this review? What are the new contributions

Considering the reviewer's observations, we have added the question RQ7: “Which groups are most affected by poverty, inequality and climate change, according to the articles analysed?” in section 2 (Materials and methods) (Line 107).

We have also introduced a new sub-section, "3.7. Main contributions" (Lines 252-306), which specifies, on the one hand, the relationship between poverty, inequality and climate change in the articles analysed and, on the other hand, the groups and sectors most affected by it, as well as other findings related to Covid-19 and the SDGs.

About the other issues raised by the reviewer in this section:

  • Why is the timeline of 1999 to 2020 chosen?

According to the Wos database, we chose this period because 1999 was when the first publication appeared. We began this work in February 2021. At this date, we could not consider the analysis of the whole year, so we chose not to include it.

  • Why is COVID-19 literature not included?

Considering your comments on Covid-19, we have added a section on the effects of covid-19 in the introduction. About the systematic review of the literature and following your question, we have analysed the articles published in 2020 to find out how many of them refer to covid-19 within the subject analysed. We present these results in section 3.7. We have also included comments in the discussion and conclusions section, also shown below.  

13.-The discussion section reads like an extended literature review than an organized discussion of the findings from the review of 201 publications.

This section aims to provide a discussion on the relationship between poverty, inequality and climate change. The conclusions provide a summary of the most relevant aspects of the SLR. In the discussion section, we highlight the relationship between these concepts, based on the results obtained in section 3.7 and the role of the circular economy as a possible contribution to improving this problem. In addition, we have referred to the SDGs and COVID-19. However, we have deleted this paragraph “To answer the first question, based on the results obtained in this research, we can af-firm that there is a close relationship between climate change and inequality since cli-mate-induced effects negatively impact health, gender, and income inequality, among others, as shown below” with “To resolve the first question and based on the results obtained in this research, we can affirm a close and direct relationship between the 3 SDGs analysed, as 73.03% of the works analysed demonstrate this. Furthermore, we can see that the effects of climate change generate patterns of inequality in health, gender and income, as indicated below” (Lines 315-318).

We have also added lines 393-402.

14.-The discussion of circular economy is opined by authors rather than the findings from the review. Some of the explanation about the circular economy and agriculture does not make clear sense. For example, how the farmers’ interactions with the market and commercialization make them vulnerable and how geographic and household differentiations make them vulnerable. It is unclear.

Following the reviewer's indications, we have changed the expression "In the case of farmers occupying these territories, in-creased market interaction and commercialisation could cause them serious adaptation problems unless their access to the technical knowledge, inputs, land and natural resources needed for agricultural and non-farm activities is improved" with " This is the case of the farmers who occupy these territories. They should adapt to market needs by introducing technical improvements and natural resources that do not damage the ecosystem so as not to remain on the margins of society or generate patterns of social vulnerability [39, 66]”. (Lines 384-387).

Reviewer 2 Report

At your result part (P4-p9), you should have revealed research evidences on the relationship between climate change, poverty and inequality. We don't see any concrete analysis on that. And based on your findings,  you should have logical discussion on the relationship between climate change, poverty and inequality. 

Author Response

Reviewer 2:

We want to thank the reviewer for his/her comments and suggestions about the work previously entitled “Analysis of research on the SDGs: The relationship between climate change, poverty and inequality". These modifications undoubtedly improve the quality and accuracy of the article.

We have highlighted all changes corresponding to such suggestions in red colour fonts in our manuscript.

Following your comment: “At your result part (P4-p9), you should have revealed research evidences on the relationship between climate change, poverty and inequality. We don't see any concrete analysis on that. And based on your findings, you should have logical discussion on the relationship between climate change, poverty and inequality”, we have introduced a section, entitled "3.7. Main contributions", to better connect with the discussions of this research. (Lines 252-306)

Round 2

Reviewer 1 Report

The authors have sufficiently addressed my comments. There are some minor issues with writing. Professional proofreading and edits are required before the publication.

Reviewer 2 Report

The authors conducted analytical and logical reviews of existing literature of SDGs related to the relationship between climate change, poverty, and inequality. This paper will make a significant contribution to this subject. However, the effects of the COVID-19 pandemic on this issue should be verified with more evidence, although authors introduced recent literature on COVID-19 and its effects on climate change, poverty, and inequality. Authors do not need to describe the effects of COVID-19 too much without ample evidence (Introduction, discussion, and conclusion parts).